# Integrated Photonic Lattice Filter for Accelerating Deep Convolutional Networks

**Matthew J. Filipovich**
University of Oxford

**Folkert Horst**
IBM Research

**Bert Jan Offrein**
IBM Research

## Abstract

The increasing computational demands of machine learning models have driven interest in developing unconventional computing hardware to improve speed and energy efficiency. In this work, we introduce an integrated photonic chip designed to perform convolution operations in deep convolutional neural networks. The convolutional kernel is implemented in the optical circuit, which functions as a two-port lattice filter, by modifying the optical signal paths through phase shifters. Using a simulated model of the optical chip that implements the convolutional layers, we evaluate the performance of a deep convolutional network trained on the CIFAR-10 dataset. We also examine the impact of hardware limitations, such as system noise and quantization, on the model performance.

## 1   Introduction

Convolutional neural networks (CNNs) are fundamental deep learning architectures widely used in computer vision tasks such as image classification, object detection, and video analysis [1, 2]. Although "state-of-the-art" CNN architectures continue to evolve, convolutions remain the primary operation executed by these models. As digital computers near their fundamental energy and speed limitations, there is a significant need for novel hardware capable of supporting the growing computational requirements of deep learning models.

Integrated photonics is a promising platform for implementing machine learning systems due to its potential for high throughput and low power consumption [3]. Neural network inference and training have both been demonstrated using photonic integrated chips [4, 5, 6]. Photonic circuits for implementing small-scale CNNs have also been proposed and experimentally validated [7, 8]. Several methods for encoding neural network weights in optical systems have been proposed, including Mach-Zehnder interformeters (MZIs) [9], microring resonators [10], and phase-change materials [11].

In this paper, we design and simulate an integrated photonic circuit capable of executing convolution operations for application in deep CNNs. We present a method for implementing 2D multi-channel convolutions using the optical circuit and model the performance of the optical hardware. We simulate a deep CNN based on the VGG-11 model, with convolutional layers implemented using the optical circuit, and train the model on the CIFAR-10 dataset [12, 13]. Finally, we evaluate the impact of phase shifter and measurement noise on the model accuracy, as well as the effect of quantization due to conversion between the analog and digital domains.

## 2   Integrated Photonic Circuit

In this section, we introduce the silicon photonic chip, shown in Fig. 1, designed to perform convolution operations. The optical circuit implements a finite impulse response (FIR) filter using optical delay lines and tunable MZIs [14]. This design enables the convolution of a time-discrete input

38th Second Workshop on Machine Learning with New Compute Paradigms at NeurIPS 2024(MLNCP 2024).

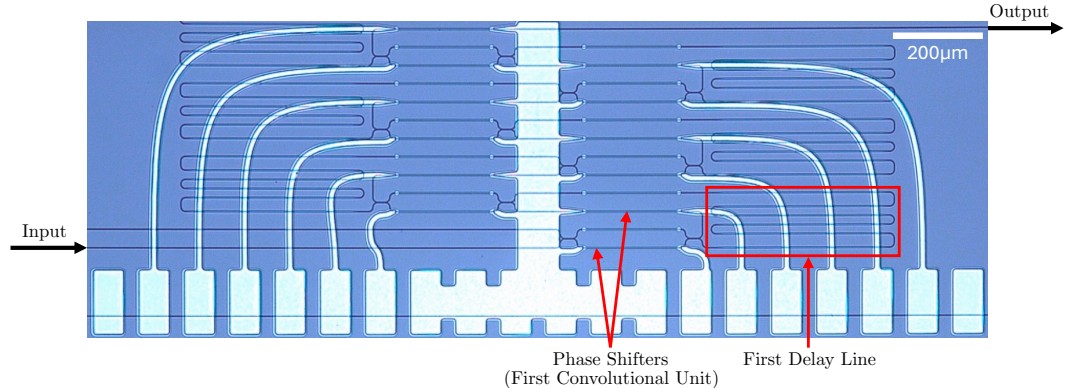

Figure 1: **Silicon photonic chip with six convolutional units.** The optical circuit implements an FIR filter to perform convolution operations. The convolutional kernel is determined by the impulse response of the system, which can be tuned using thermo-optic phase shifters. The waveguides, designed for a wavelength of $1.55\,\mu\text{m}$, are $500\,\text{nm}$ wide and $220\,\text{nm}$ thick.

signal with a finite convolutional kernel at each time step, separated by an interval of $\Delta t$:

$$z_k = g\left(\sum_{n=1}^{N} h_n\, x_{k-n}\right). \tag{1}$$

Here, $x_k$ and $z_k$ are the input and output signals at the $k$-th time step, respectively; $h_n$ is the $n$-th element of the convolutional kernel, and $g(\cdot)$ is the measurement function applied by the photodetection setup. The input optical signals are coherent, which enables both phase and amplitude modulation. The complex-valued kernel elements $\{h_1, \ldots, h_N\}$ are determined by the impulse response of the optical system and can be modified using thermo-optic phase shifters. Depending on the photodetection setup, the output measurement can yield either the complex-valued field or the optical intensity.

At each time step, the next input signal is fed into the optical circuit and the output is measured. This process is equivalent to shifting the convolutional kernel across the input by one position. By reusing input signals already propagating through the optical circuit, the system performs $N$ multiply-accumulate operations at each time step for a convolutional kernel with $N$ elements. Over $T$ time steps (for an input of size $T$), the resulting convolution output has size $T - N + 1$.

The optical circuit is composed of a sequence of *convolutional units*, each containing a delay line and tunable MZI, as illustrated in Fig. 2. The MZI can be modeled as a two-port device containing two 3 dB directional couplers connected by two waveguides. The relative phase between the two paths can be adjusted using the phase shifter $\varphi$, which determines the amount of light directed to the top and bottom output ports. The top output port of the MZI is linked to a phase shifter $\theta$, while the bottom port is connected to a delay line (a folded waveguide) that introduces a propagation delay of $\Delta t$.

Using an input signal with frequency $\omega$ and treating the optical circuit as lossless, the scattering matrix of the $c$-th convolutional unit is

$$S^{(c)} = \frac{1}{2}\begin{pmatrix} e^{i\theta_c} & 0 \\ 0 & e^{i\omega\Delta t} \end{pmatrix}\begin{pmatrix} 1 & i \\ i & 1 \end{pmatrix}\begin{pmatrix} 1 & 0 \\ 0 & e^{i\varphi_c} \end{pmatrix}\begin{pmatrix} 1 & i \\ i & 1 \end{pmatrix}, \tag{2}$$

which simplifies to

$$S^{(c)} = \frac{1}{2}\begin{pmatrix} e^{i\theta_c}(1 - e^{i\varphi_c}) & ie^{i\theta_c}(1 + e^{i\varphi_c}) \\ ie^{i\omega\Delta t}(1 + e^{i\varphi_c}) & e^{i\omega\Delta t}(-1 + e^{i\varphi_c}) \end{pmatrix}. \tag{3}$$

The impulse response of the system depends on the scattering matrix of each convolutional unit. Specifically, the $n$-th element of the convolutional kernel, $h_n$, is determined by the superposition of optical paths with a total propagation delay of $n\Delta t$ resulting from the delay lines. For a circuit

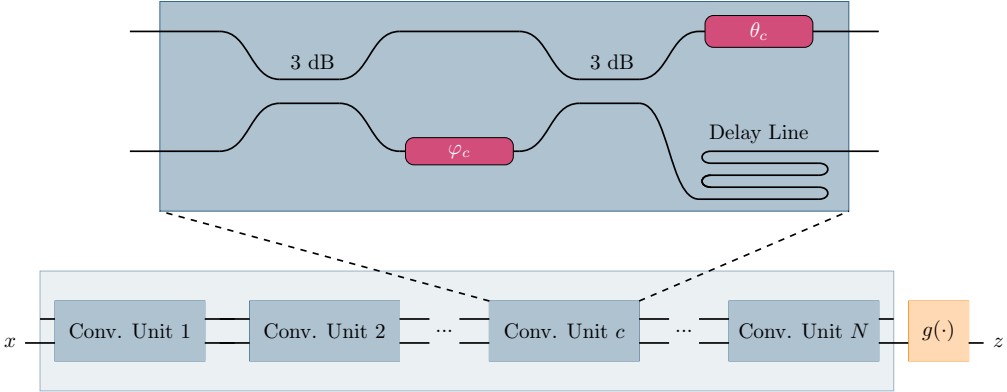

Figure 2: **Optical circuit schematic.** The optical circuit (*bottom*), modeled as a two-port lattice filter, consists of a sequence of $N$ convolutional units, followed by the measurement function $g(\cdot)$. Each convolutional unit (*top*) includes a delay line and tunable MZI with phase shifters $\varphi$ and $\theta$.

composed of $N$ convolutional units, this is given by

$$h_n = \sum_{p \in \mathcal{P}_{n\Delta t}} \left( \prod_{c=1}^{N} S_{i_p, j_p}^{(c)} \right),$$
(4)

where $p$ represents a path from the set of all optical paths $\mathcal{P}_{n\Delta t}$ with total propagation delay $n\Delta t$, and $S_{i_p, j_p}^{(c)}$ is the $(i_p, j_p)$ entry of the scattering matrix defined in Eq. (3).

For example, the convolutional kernel elements for a circuit consisting of three convolutional units are defined as

$$
\begin{aligned}
h_1 &= S_{2,1}^{(3)} \cdot S_{1,1}^{(2)} \cdot S_{1,2}^{(1)}, \\
h_2 &= S_{2,2}^{(3)} \cdot S_{2,1}^{(2)} \cdot S_{1,2}^{(1)} + S_{2,1}^{(3)} \cdot S_{1,2}^{(2)} \cdot S_{2,2}^{(1)}, \\
h_3 &= S_{2,2}^{(3)} \cdot S_{2,2}^{(2)} \cdot S_{2,2}^{(1)}.
\end{aligned}
$$
(5)

An illustration of the optical path required to implement a simple kernel is shown in Fig. 3.

**2D Multi-Channel Convolutions**  The previously described approach can be generalized to perform 2D multi-channel convolution operations using the optical circuit. In this method, each row of an $M \times N$ filter is treated as an independent 1D filter applied sequentially across the image in a series of 1D convolutions. Convolutions across several input channels are executed by processing each channel individually and then summing the outputs at each spatial pixel.

In 2D multi-channel convolutions, the kernel is represented as a 4D tensor $\mathbf{H}$, where $H_{i,l,j,k}$ is the kernel element corresponding to the $i$-th output channel, $l$-th input channel, $j$-th spatial row, and $k$-th spatial column. The 3D input and output tensors are denoted as $\mathbf{X}$ and $\mathbf{Z}$, respectively, where $\mathbf{Z}$ is the output from convolving $\mathbf{H}$ across $\mathbf{X}$:

$$\mathsf{Z}_{i,j,k} = \sum_{l=1}^{L} \sum_{m=1}^{M} g \left( \sum_{n=1}^{N} \mathsf{H}_{i,l,m,n} \, \mathsf{X}_{l,j-m,k-n} \right),$$
(6)

over all valid tensor index values of $i$, $j$, and $k$ [15]. As in Eq. (1), $g(\cdot)$ denotes the measurement function applied by the photodetection setup. Similar to the 1D case, the values for $\mathbf{H}$ are determined by the impulse response of the optical circuit.

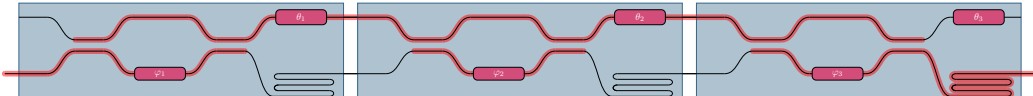

Figure 3: **Optical path for a specified convolutional kernel.** The optical path, illustrated in red, required to implement the simple kernel $h_1 = 1$, $h_2 = 0$, and $h_3 = 0$ in a circuit consisting of three convolutional units. In this case, the signal is only delayed by one time step by routing through the final delay line in the third convolutional unit.

## 3  Results

**Optical Convolutional Layer**  We simulated a 2D convolutional layer using PyTorch that models the optical chip using Eqs. (4) and (6) [16]. During training, instead of optimizing the convolutional kernel **H** of each layer directly, we treat the set of phase shifters $\{\varphi_1, \ldots, \varphi_N, \theta_1, \ldots, \theta_N\}$ as learnable parameters. This approach accurately models the optical circuit and accounts for constraints imposed by the hardware.

Our model accounts for imperfections in the optical system by injecting noise and quantizing values where necessary. We implement the following features in our simulation:

- **Phase shifter noise:** Gaussian noise with standard deviation $\sigma_{\varphi,\theta}$ is added to the phase shifter values during each forward pass.

- **Measurement Noise and Quantization:** We use a coherent detection setup to measure the real-part of the complex-valued output signal. The measurement function $g(\cdot)$ includes Gaussian noise from the transimpedance amplifier (which converts current to voltage), followed by bit quantization performed by the analog-to-digital converter. Quantizing to $N_b$ bits, the noise has a standard deviation of $2/2^{N_b}$.

- **Input Signal Quantization:** The input signal is quantized to $N_b$ bits by the digital-to-analog converter.

**CIFAR-10 Training**  We implemented a deep CNN model in PyTorch using the previously discussed optical convolutional layers. The model, shown in Fig. 4, is based on the VGG-11 architecture and contains 18.4 million parameters [13]. The Tanh activation function is used in the model to ensure the output values are within a known range. We employed the Adam optimizer with a learning rate that followed a cosine decay schedule from 0.01 over 200 epochs. The model was trained with an augmented CIFAR-10 dataset, using random cropping and horizontal flipping.

We trained several CNNs using varying levels of noise added to the phase shifters. This enabled evaluation under different hardware conditions, leading to insights regarding the hardware specifications required to achieve the desired performance. We evaluated the CNNs on the test dataset using quantization levels $N_b$ of 8, 6, and 4 bits. The training and validation curves during training are shown in Fig. 5, and the test results under different quantization conditions are given in Table 1.

Table 1: Test accuracy (%) achieved by models trained on CIFAR-10 dataset with quantization to $N_b$ bits.

| $\sigma_{\varphi,\theta}/2\pi$ | $N_b = 8$ | $N_b = 6$ | $N_b = 4$ |
|---|---|---|---|
| $2^{-7}$ | 89.14 | 88.70 | 73.64 |
| $2^{-6}$ | 88.87 | 88.62 | 72.89 |
| $2^{-5}$ | 87.22 | 86.99 | 76.57 |
| $2^{-4}$ | 81.90 | 81.74 | 74.25 |
| $2^{-3}$ | 62.04 | 61.28 | 50.70 |

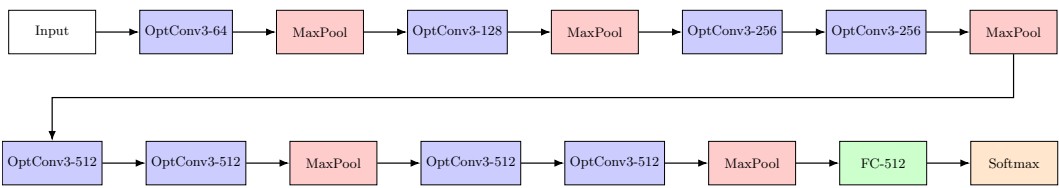

Figure 4: **Convolutional network model based on VGG-11.** The optical convolutional layers, denoted as "OptConv[kernel size]-[output channels]," are implemented using the simulated optical circuit. Each convolutional layer is followed by BatchNorm and a Tanh activation function, which are not shown.

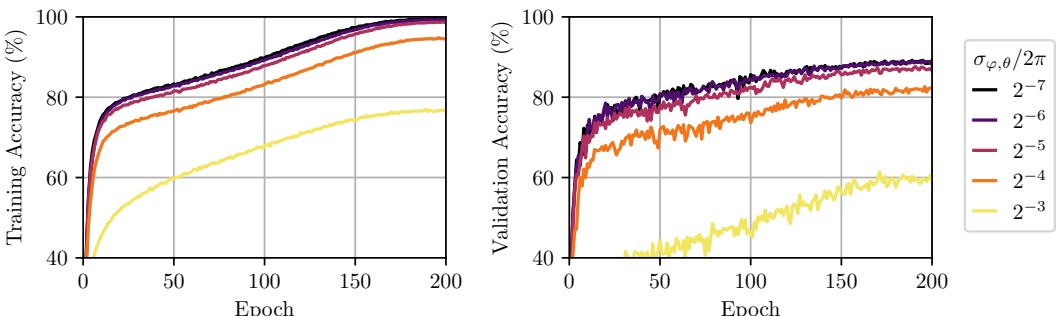

Figure 5: **Model performance during training.** Training accuracy (*left*) and validation accuracy (*right*) of models trained on the CIFAR-10 dataset. Models have varying levels of phase shifter noise defined by $\sigma_{\varphi,\theta}$.

## 4   Discussion

Integrated photonic chips offer several advantages for information processing tasks, including the ability for real-time processing of high-speed data [17]. One of the primary advantages of our proposed photonic circuit is the ability to reuse the input signal over multiple convolution operations. This feature can enhance the system's overall energy efficiency and throughput speed, as a circuit with $N$ convolutional units can perform up to $N$ multiply-accumulate operations at each time step while only feeding in a single input value.

The simulation results demonstrate that the photonic circuit is capable of successfully implementing convolutional layers in deep CNNs. As expected, the performance of the model is dependent on the amount of noise present in the system, as well as the level of quantization. The model performance generally improves with reduced system noise and increased bit precision. The CNN trained with the lowest noise ($\sigma_{\varphi,\theta}/2\pi = 2^{-7}$) achieved the highest test accuracy of 89.14% using $N_b = 8$ bits. This performance is similar to the accuracy attained by a model we trained using the same architecture with standard convolutional layers (i.e., floating-point arithmetic), which achieved a test accuracy of 89.65% without quantization.

A crucial consideration for the design of analog neural networks is the impact of noise in the system. Several mitigation strategies have been proposed depending on the source of the noise [18]. Additionally, the inclusion of noise during the training process has shown to improve system robustness and reduce the simulation-to-reality gap [19]. As shown in Table 1, the model accuracy generally decreases as the phase shifter noise increases. However, at low precision of $N_b = 4$ bits, the best performance is achieved by the CNN trained with $\sigma_{\varphi,\theta}/2\pi = 2^{-5}$. In this case, we believe the models trained with less noise ($2^{-6}$ and $2^{-7}$) perform worse during testing because they are less resilient to imperfections in the system.

The current model architecture uses the Tanh activation function, which ensures the values passed to succeeding layers are within a finite range. This is necessary for implementation in analog hardware, as the minimum and maximum values of the input are required to encode the data onto optical signals. Implementing non-linear activation functions without finite bounds, such as the ReLU

function, may require normalization techniques during training to transform the range of values to a finite scale.

In future work, we aim to experimentally demonstrate the use of our photonic integrated chip for implementing a deep CNN. We will use our simulation to determine the optimized values for the phase-shifters, which will be applied to the optical circuit for model evaluation. The simulation can be further improved by incorporating experimental data from the photonic hardware during the training process. This approach, known as "hardware-in-the-loop training," has been shown to mitigate the impact of hardware imperfections on system performance [20]. The application of in-situ training, where the training process is performed directly on the photonic chip rather than using a digital computer, can also be explored as a method to improve results.

## Acknowledgements

The authors acknowledge funding from the European Union's Horizon 2020 research and innovation program under the grant agreements No. 101017237 (PHOENICS), No. 101070195 (PROMETHEUS), and No. 956071 (AppQInfo).

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
