# OpenReview forum: "Integrated Photonic Lattice Filter for Accelerating Deep Convolutional Networks"
_NeurIPS.cc/2024/Workshop/MLNCP — MLNCP Poster_

### Official Review · Reviewer_8Fmc · 2024-10-04
**This paper presents a silicon photonic circuit for 1D convolutions applied over sequential inputs in discrete-time. The circuit is simulated in software and its parameters are optimized for CNNs trained with pytorch. The effect of noise is studied. Unfortunately, performance estimations of the real hardware are missing.**

**Rating:** 6
**Confidence:** 3

**Review:**

The authors present a novel silicon photonic circuit for 1D convolution over input signals provided in discrete time steps. A photograph of the circuit is shown, but no details about a hardware implementation are given (like chip size etc.). The circuit uses Mach-Zehnder-Interferometers, which is a common approach in neuromorphic photonics. The novelty is the proposal of a convolution circuit unit which, when chained, allows for N-coefficient 1D convolutions. The circuit’s transfer function is theoretically described by means of scattering matrices, its overall functionally is also nicely described in text and illustrative figures.
Next, the convolution circuit and its parameters are integrated into pytorch, and a VGG-11 architecture is trained on CIFAR-10. By this approach, the circuit parameters are directly trained, and no conversion from normal convolutional layers is needed (avoiding any degradation after conversion).
Different noise levels of the phase shifters and different quantization levels of inputs and outputs are studied. The model with lowest noise and 8-bits even achieves a higher test accuracy than the full-precision floating-point reference CNN.
The main contributions of this work are the circuit concept and the methodology for training the CNN with such circuits.
More details about the hardware (technical details, circuit measurements or simulations) and the concept for integration of this circuit into a larger neuromorphic photonic chip with activation functions and multiple layers are missing but would help to understand general usability and effectiveness of the approach.

# quality
The authors present a new neuromorphic photonic circuit for 1D convolutions. This is a very nice concept, especially as it allows for direct training in frameworks such as pytorch. When feeding in a full sequence of inputs, this approach performs the equivalent number of MAC operations like the kernel size. Training results are shown for a standard CNN architecture on CIFAR-10.

# clarity
The paper is well written in general.

Here are minor suggestions for improvement:
- Fig 1: please annotate delay line, MZI, and phase shifters in the chip photo. Also provide the circuit dimensions
- The capability of the circuit, that you can feed in an L-size input signal and process it in L timesteps to obtain an (L-K+1) size output signal (assuming K is the kernel size), could be better explained. It was at the beginning unclear to me, whether for the 2nd output value, we need to re-feed the input or not.

# originality
The authors provide a novel circuit architecture for 1D convolution. This is accompanied with a direct training of the circuit model parameters, which is expected to be superior to conversion of normally trained convolutions.
significance
The silicon photonic circuit offers a new way to implement 1 D convolutions. Whether this solution offers better performance (speed, energy, robustness to noise) than other solutions will only become clear when a real-chip has been produced and measured. Also, its integrability into a full neuromorphic photonic system has yet to be discussed.

# pros
- authors suggest a novel silicon photonic circuit for 1D convolution
- its functionality is well described in text and in equations
- the circuit’s function is replicated in pytorch and the circuit parameters (phase shifts \theta and \phi) are directly trained
- For a low noise-level the approach achieves accuracies very close to reference CNN with same architecture and full-precision floating point weights.

# cons
- No concept is presented how this circuit shall be integrated into a larger neuromorphic photonic system, both in terms of structure and in terms of operation (Is the approach cascadable?)
- Suggested concept relies on D/A and A/D converters, which potentially consume a lot of power. Especially in case of 2D convolutions, the digital outputs of the circuit represent partial sums of the Conv2D which need to be accumulated. Hence, digital buffers are needed.
- Details about the silicon photonic circuit missing: size, technology, noise, energy estimation, lowest \Delta_t in practice
- Comparison to other work doing photonic convolutions is very brief. Remaining space of the paper could be used for a more detailed discussion.

---

### Decision · Program_Chairs · 2024-10-10

Accept (Poster)